# IAMNN: ITERATIVE AND ADAPTIVE MOBILE NEURAL NETWORK FOR EFFICIENT IMAGE CLASSIFICATION

**Sam Leroux**[*]**, Pieter Simoens & Bart Dhoedt**
Ghent University - imec, IDLab
Department of Information Technology
B-9052 Ghent, Belgium

**Pavlo Molchanov, Thomas Breuel & Jan Kautz**
NVIDIA
USA

## ABSTRACT

Deep residual networks (ResNets) made a recent breakthrough in deep learning. The core idea of ResNets is to have shortcut connections between layers that allow the network to be much deeper while still being easy to optimize avoiding vanishing gradients. These shortcut connections have interesting properties that make ResNets behave differently from other typical network architectures. In this work we use these properties to design a network based on a ResNet but with parameter sharing and adaptive computation time. The resulting network is much smaller than the original network and can adapt the computational cost to the complexity of the input image.

## 1 INTRODUCTION AND RELATED WORK

After their impressive results on the ILSVRC2015 challenge, deep residual networks (He et al., 2016) quickly became one of the default architectures for computer vision tasks. Instead of just stacking layers on top of each other where each layer has to transform the output of the previous layer ($h_{i+1} = F_i(h_i)$), they add *skip connections*, identity mappings that copy the input of the layer to the output. The layer then learns a *residual* to add to the input ($h_{i+1} = h_i + F_i(h_i)$). ResNets are closely related to Highway networks (Srivastava et al., 2015) where the flow of information is regulated by gates instead of using a fixed skip connection.

The residual connections have some very interesting properties. Various works have shown that ResNets are remarkably robust against deleting or reordering layers from a trained network (Veit et al., 2016) (Srivastava et al., 2015), while this behavior is not found in the more traditional architectures. Veit et al. (2016) argue that ResNets should be interpreted as an exponential ensemble of shallower models. Greff et al. (2016) argue that the layers in a ResNet do not learn completely new representations but instead they gradually refine the features extracted by the previous layers. In this *unrolled iterative estimation view* successive layers cooperate to compute a single level of representation. Jastrzebski et al. (2017) provided a formal view of iterative feature refinement and showed that a each residual layer refines the feature representation to reduce the loss with respect to the hidden representation.

In this work we follow the iterative estimation view and exploit these properties to design networks with less parameters and a smaller computational cost. To reduce the model size we enforce a strict parameter sharing between the layers in the network. Since each layer refines the features extracted by the previous layer instead of learning completely new features, it seems plausible that there is a certain redundancy in having different parameters for each layer. Sharing parameters between layers has been proposed before (Jastrzebski et al., 2017) (Boulch, 2017) and both papers report impressive model size reductions.

To reduce the computational cost we incorporate an Adaptive Computation Time (ACT) mechanism. In a traditional ResNet every sample follows the exact same path through the network where every layer refines features. Not all samples are equally hard to classify and some of them will need more refinement steps than others. With ACT we can include a component that will decide how many steps are needed. ACT was introduced by Graves (2016) for recurrent neural networks where the

---

[*]The research work was done during an internship at NVIDIA, Santa Clara, California, USA

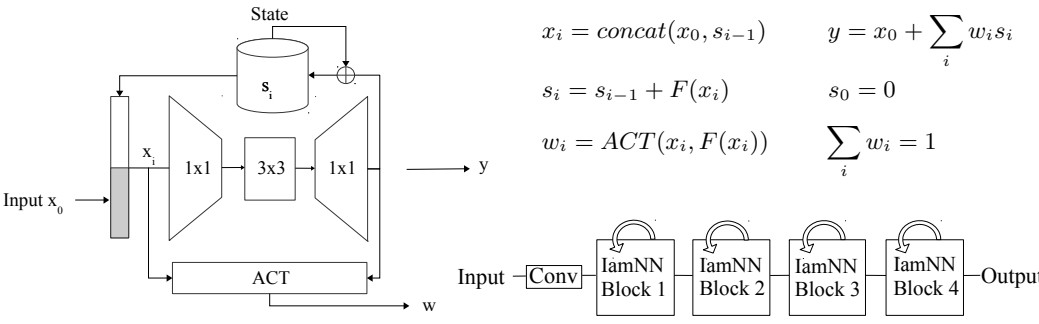

$$x_i = concat(x_0, s_{i-1}) \qquad y = x_0 + \sum_i w_i s_i$$

$$s_i = s_{i-1} + F(x_i) \qquad s_0 = 0$$

$$w_i = ACT(x_i, F(x_i)) \qquad \sum_i w_i = 1$$

Figure 1: The IamNN network has the same structure as a ResNet but multiple residual units in each block are replaced with the architecture on the left which reuses the same weights multiple times.

network adapts the number of calculation steps to the complexity of the input. ACT can also be applied to networks for image recognition. Figurnov et al. (2016) introduced a Spatial ACT for ResNet which adapts the amount of computation for spatial locations.

## 2 ARCHITECTURE

A typical ResNet starts with a convolutional layer with batch normalization and ReLu nonlinearity followed by maxpooling. Then, a sequence of four blocks is stacked where each block consists of multiple stacked residual units. Each residual unit consists of one or more convolutional layers and a shortcut connection. The first convolutional layer in each block uses a convolution with stride 2 to reduce the spatial size.

We propose to replace each block by the architecture shown in Figure 1 (left). Each residual unit in a block is replaced by an iteration of this module. We use maxpooling between each block to reduce the spatial size. Our block consist of three main parts: a processing block with three convolutional layers, a state buffer where the results are accumulated and the ACT block that decides how many iterations of this block are needed. The state buffer is used to gradually build the output of the block. The output of each iteration is added to the state which allows for an iterative refinement of the features. The initial state ($s_0$) is initialized with zero values.

The processing block consists of three convolutional layers with a bottleneck structure (He et al., 2016). At each iteration we concatenate the original input of the block ($x_0$) with the current state ($s_{i-1}$) before passing the concatenated vector through the processing block. The output of the processing block is added to the state. Each convolutional layer is followed by batch normalization. We found that it is important to use a different set of statistics in each iteration for batchnorm. This only incurs a small overhead since the number of batchnorm statistics is very small compared to the full network. The same approach was used in Jastrzebski et al. (2017) in their experiments.

The ACT block is a small two-layer fully connected network (2 times 64 hidden units) that decides whether to keep evaluating the block or to move on to the next block. We concatenate the current state, the original input and the output of the processing block and apply global average pooling to obtain a vector with a single value for each channel. The ACT network uses this vector to calculate a *halting score* between 0 and 1 (sigmoidal activation). We sum the halting scores of each iteration and as soon as the cumulative halting score reaches one we stop evaluating this block and move on to the next block. We add an additional loss term to encourage the network to use fewer iterations.

## 3 RESULTS

We trained our models on three datasets for image recognition: CIFAR10, CIFAR100 and ImageNet. We report the number of parameters, the theoretical number of operations and the test accuracy in Table 3. We designed our models after a ResNet architecture and constrained them to use at maximum the same number of iterations in each block as the the number of units in that block. For example IamNN based on ResNet152 is constrained to use at maximum 3, 8, 36 and 3 iterations for each block respectively. The computational cost of the IamNN networks varies depending on the complexity of the input image, we report the average required FLOPS over all images in the test set.

For CIFAR10 and CIFAR100 we reduce the number of parameters by 90% compared to the corresponding ResNet101. The number of FLOPS varies between $0.8 - 2.0$ GFLOPs, depending on the

easy images                                                                              hard images

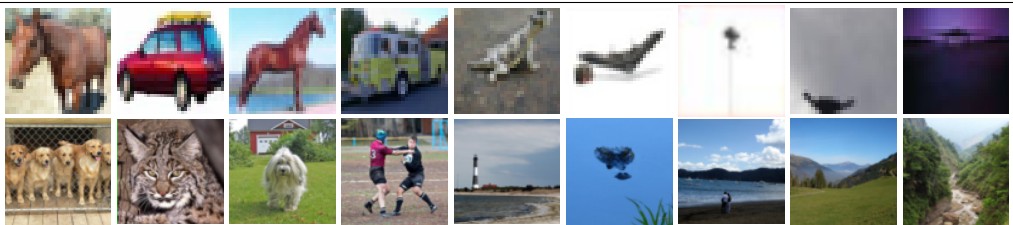

Figure 3: The spectrum of easy (left) to hard (right) to classify images for the network trained on CI-FAR10 (top row) and ImageNet (bottom row). The network automatically adapts the computational cost to the complexity of the input image.

complexity on the input image. On average we require 1.1G and 1.6G operations for CIFAR10 and CIFAR100 respectively, a reduction of 56% and 36% respectively. For CIFAR10 we obtain a slightly higher accuracy compared to the ResNet, probably because weight sharing acts as a regularizer for the small dataset. On CIFAR100 however the accuracy drops from 79.3% to 77.8%.

For the ImageNet dataset we used ResNet152 as a baseline. We are again able to reduce the number of parameters by 90% and the computations by 65%; the top5 accuracy drops from 93.3% to 89%. To illustrate the benefit of iterative refinement using the same weights, we also include the result when our IamNN network is only allowed one iteration per block. In this case there is no iterative refinement possible and the network obtains a top5 accuracy of 83.2%. We also compare to other architectures that were built to be very efficient in terms of computational cost and memory footprint (MobileNet (Howard et al., 2017), ShuffleNet (Zhang et al., 2017) and GoogleNet (Szegedy et al., 2014)). These networks obtain a similar accuracy with a similar number of parameters but at a lower computational cost. This is because we did not change the basic operations of the network (1x1 and 3x3 convolutions). Some of the techniques used in the efficient networks such as depthwise separable convolutions are however orthogonal to our weight sharing and ACT approach and could be used to reduce the computational cost even further.

Figure 2 shows how many iterations are used for each block in the ImageNet network. The vertical line indicates the number of residual units in the corresponding ResNet (= the maximum number of iterations for the IamNN). Figure 3 illustrates how the network is able to adapt the computational cost to the complexity of the image. We show some typical images ranked from fast (left) to slow (right) computation. The computation cost varies between 0.7G and 2G FLOPS for CIFAR10 and between 2.5G and 9G FLOPS for ImageNet.

| Dataset | Network | Params | FLOPS | Top1/ Top5 (%) |
|---------|---------|--------|-------|----------------|
| CIFAR10 | ResNet101 | 42 M | 2.5G | 93.8 |
|         | IamNN | 4.5 M | 1.1G (.7G - 2G) | 94.6 |
| CIFAR100 | ResNet101 | 43 M | 2.5G | 79.3 |
|          | IamNN | 4.6 M | 1.6G (.7G - 2G) | 77.8 |
| ImageNet | ResNet152 | 60 M | 11.5 G | 77.0 / 93.3 |
|          | ResNet18 | 12 M | 1.8 G | 69.5 / 89.2 |
| Single crop | IamNN 1 iter | 4.8 M | 0.9 G | 60.8 / 83.2 |
| Single network | **IamNN** | **5 M** | **4 G (2.5G - 9G)** | **69.6 / 89.0** |
|          | ShaResNet34 | 14 M | 11 G | 71.0 / 91.5 |
|          | Googlenet | 7 M | 1.6 G | 65.8 / 87.1 |
|          | MobileNet1 | 4.2 M | 570 M | 70.6 / 89.5 |
|          | ShuffleNet2x | 5.6 M | 524 M | 70.9 / 89.8 |

Table 1: Number of parameters and classification accuracy on three benchmark datasets for our IamNN architecture and the corresponding ResNet architecture.

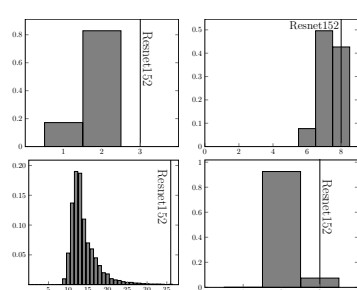

Figure 2: Number of iterations used in each of the four blocks (ImageNet network)

## 4    CONCLUSION AND FUTURE WORK

We proposed a new ResNet based architecture based on insights in the iterative refinement behavior of ResNets and Highway networks. By using the same set of weights multiple times we can reduce the model size by 90%. With adaptive computation time the computation cost of the model depends on the complexity of the input image and is on average much lower than a typical ResNets. In future work, we will incorporate other techniques such as depthwise separable convolutions to reduce the computational cost even further.

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
