# OpenReview forum: "IamNN: Iterative and Adaptive Mobile Neural Network for efficient image classification"
_ICLR.cc/2018/Workshop — Accept_

### Official Review · AnonReviewer1 · 2018-03-09
**Results are not so good**

**Rating:** 6
**Confidence:** 4

**Review:**

Enhancing deep CNN model efficiency is very important. This paper combines two existing techniques (sharing weights across layers) and ACT to develop a new architecture for ResNet, in a nice way. The presented idea and architecture are interesting. But the evaluation results are not so impressive. It is slightly better than GoogleNet but not so good as ShuffleNet and MobileNet. As discussed by the authors, the model can be further compressed using separate convolutions. It will be interesting to see such results.

---

### Official Review · AnonReviewer3 · 2018-03-09
**Important research direction; further careful analysis needed**

**Rating:** 7
**Confidence:** 5

**Review:**

This paper proposes a method of training neural networks with:
a) fewer free parameters (weights), and,
b) variable cost of inference
compared to an original Highway or Residual network. The cost of inference is restricted to be less than or equal to the original network.

The method is a natural consequence of the iterative estimation view of Highway and Residual networks put forth by Greff et al., so it only applies to networks that conform to this view. The reduction in parameters is achieved by tying the weights of multiple layers/blocks, and the variable number of feature estimation steps is learned using ACT proposed by Graves.

This is a very natural marriage of recent ideas and has practical significance. However, the results are certainly preliminary.
The authors use a very large ResNet101 as baseline for CIFAR datasets (>40M Params), when much smaller deep networks also perform very well on them. Does the method only appear to work well when the baseline network uses too many parameters/FLOPS to begin with? Is there a benefit over simply training a smaller network?
Nevertheless, the preliminary results seem sufficient for a workshop acceptance. I hope that the authors will be careful in discussing the implications of the current results.

I should also point out that Adaptive Computation in general is not completely new, additional references are mentioned by Graves (2016).

Pros:
- important direction of model development
- encouraging initial results combining ACT and iterative estimation

Cons:
- important comparisons are missing, eventual final results may not be convincing

---

### Decision · Program_Chairs · 2018-03-20
**ICLR 2018 Workshop Acceptance Decision**

**Decision:**

Accept

**Comment:**

Congratulations, your paper was accepted to the ICLR workshop.